# Fluorescence Characteristics of Dissolved Organic Matter (DOM) in Percolation Water and Lateral Seepage Affected by Soil Solution (S-S) in a Lysimeter Test

**DOI:** 10.3390/s19184016

**Published:** 2019-09-17

**Authors:** Teng-Pao Chiu, Wei-Shiang Huang, Ting-Chien Chen, Yi-Lung Yeh

**Affiliations:** 1Department of Civil Engineering, National Pingtung University of Science and Technology, Pingtung 91201, Taiwan; 2Department of Environmental Science and Engineering, National Pingtung University of Science and Technology, Pingtung 91201, Taiwan

**Keywords:** dissolved organic matter, soil solution, lysimeter test, fluorescence spectra, optical indices

## Abstract

The composition and structure of dissolved organic matter (DOM) are sensitive indicators that guide the water infiltration process in soil. The DOM chemical composition in seepage affects river water quality and changes soil organic matter (SOM). In this lysimeter test study, fluorescence spectra and optical indices were used to examine the interaction between the percolation water (P-W) and leachate water (L-W) DOMs affected by the soil solution (S-S). The L-W DOM had a higher aromaticity (SUVA_254_), average molecular weight (S_275-295_) and terrestrial source (fluorescence index (FI)), but fewer autochthonous sources (biological index (BIX)) than the P-W DOM. Organic carbon standardization (OCS) and protein- (PLF), fulvic- (FLF) and humic-like fluorescence (HLF) intensity showed that L-W DOM increased 44%, 55% and 81%, respectively, compared to the P-W DOM. The linear regression slopes between OCS FLF and PLF were 0.62, 1.74 and 1.79 for P-W, L-W and S-S, respectively. The slopes between OCS HLF and PLF were 0.15, 0.58 and 0.64 for P-W, L-W and S-S, respectively. The P-W DOM was in contact with the soil litter layer, where S-S labile lignin phenolic compounds released and dissolved into the L-W DOM. This increased its aromaticity, and extent of humification.

## 1. Introduction

Dissolved organic matter (DOM) is an important component of water; it changes drinking water quality and reduces water treatment efficiency [1,2]. Surface water DOM infiltration into soil is affected by the natural DOM characteristics such as chemical structure, functional group content or molecular size [3]. Furthermore, DOM transport is controlled by both physicochemical and biological processes in soil that serve to retain and transform the DOM and release soil organic matter (SOM) from different soil fractions [4,5,6].

The hypothesis of DOM infiltration into the deep groundwater aquifer assumes that there are two processes controlling the DOM chemical composition and structure. One is that DOM hydrophobic components adsorb on soil minerals and/or DOM partitions into the SOM. The other process is that the DOM metabolism by microorganisms subsequently produces small molecular weight, hydrophilic and autochthonous DOM and the metabolites of microorganisms [4,6,7]. Therefore, the DOM in the deep groundwater aquifer has mainly biological sources. However, the shallow infiltration of lateral seepage in river banks and drainage channels may yield a preferential release of dissolved DOM containing soil plant litter lignin, which contains phenolic compounds [4,6,7]. The shallow seepage may increase the dissolved organic carbon (DOC) concentration and change the DOM chemical composition and structure. The discharged water DOM results in degraded river water quality and causes SOM loss [7,8].

DOM and SOM are heterogeneous and complex organic mixtures. Many advanced techniques have been applied to the qualitative and quantitative DOM and SOM chemical composition and structure, such as NMR, FTIR, HPLC and Py-GC-MS. Advanced technical analysis requires complex pretreatment processes and is time consuming [9,10,11,12,13]. Spectral methodology, UV-Vis and fluorescence spectroscopy are rapid, non-destructive and sensitive detection methods that are widely used to determine measured chemical compositions and structures of various DOM and SOM [11,14,15,16,17].

UV-Vis indices SUVA_254_ and S_275-295_ show DOM and SOM aromaticity and average molecular weight, respectively [15,17,18]. The fluorescence index (FI) distinguishes terrestrial source contribution [17,19,20]. The biological Iindex (BIX) distinguishes the contribution of autochthonous sources [17,21]. A fluorescence excitation/emission matrix (EEM) analyzes DOM species and water quality characteristics with fluorescent intensity and intensity ratios [22,23,24,25]. The three regions of protein- (PLF), fulvic- (FLF) and humic-like fluorescence (HLF) represent the wavelength ranges of intensities in EEM, respectively. The chemical composition of DOM and SOM were examined by the fluorescence intensity and ratio change in the three regions [26,27].

Previous studies have used optical indicators to explore DOM differences and mainly focused on surface water and deep groundwater [6,28,29,30]. However, research on the percolation water (P-W) DOM in shallow infiltration and lateral seepage affected by the SOM chemical composition and structure is still lacking.

In this study, using a medium-scale lysimeter, optical indices (UV-Vis and fluorescence) and fluorescence spectroscopy were used to examine the chemical structure and composition relationship between P-W and leachate water (L-W) DOM as well as extracted soil organic solution (S-S). The aromaticity, average molecular weight and the terrestrial and autochthonous contributions were used to examine the P-W and L-W DOM as well as the S-S with optical indicators SUVA_254_, S_275-295_, FI and BIX, respectively. In addition, the fluorescent intensity differences between PLF, FLF and HLF in P-W, L-W and S-S was examined with intensity mean and linear regression. The chemical structure and composition differences between P-W and L-W DOM and how they were affected by the S-S were discussed.

## 2. Research Method and Material

### 2.1. Research Site and Sample Collection

The research site is located at an agricultural experiment site in southern Taiwan. The configuration of the lysimeter and the geographical location of the experimental site is shown in Figure 1. The lysimeter is 5 m long, 3 m wide and 2 m in depth. The bottom layer (1 m) was filled with sand and gravel and the upper 1 m layer was filled with nearby farm soil. The lysimeter is surrounded by concrete. The drainage pipes that collect the L-W were placed at soil depths of 0.3 and 0.6 m. The percolation water contact time with the soil was less than 30 min.

### 2.2. Water Samples Collection

There were three types of samples in this study. The P-W sample was well water pumped from 35 m and 15 m deep groundwater away from the test site. The L-W sample was collected from the drainage pipe outlet as shown in Figure 1. The S-S sample was extracted from the collected soils in the test site. Water quality parameters (pH, dissolved oxygen (DO), EC) of P-W and L-W were measured as well as the soil basic properties, pH and soil organic matter content. Table 1 lists the water quality of P-W and L-W and soil basic properties.

The water infiltration tests were carried out four times. In each test, two P-W samples were collected (one for each lysimeter) for a total of eight P-W samples. Additionally, in each test, eight L-W samples (four samples for each lysimeter) were collected for a total of 32 L-W samples. The DOC concentration, UV-Vis and fluorescence spectra of the water samples were measured after filtration with a membrane (<0.45 μm).

### 2.3. Soil Collection and Extraction

Before each percolation test, an auger (inner diameter, 5 cm) was used to collect soil samples to a soil depth of 60 cm. Each of the soil columns was uniformly mixed to produce a soil sample. Two soil samples were taken from each lysimeter in each test; a total of 16 soil samples was collected in the four experiments. The hole was refilled with the adjacent farm soil.

The soil samples were air-dried and large particles were removed. The soil samples were then ground and sieved at 2.0 mm (sieve #10). The sieved soil sample was washed with 0.1 N HCl acid to remove alkaline earth metals and carbonate. The residual soil sample was mixed with a 0.1 N NaOH solution at a ratio of 1/20 (soil/water) and extracted with a reciprocating shaker (150 rpm, 24 h). The soil suspension was centrifuged at 3200 g for 20 min and the supernatant was passed through a 0.45 μm membrane (Pall) to collect the soil solution (S-S). The S-S solution was stored at 4 °C. The DOC concentration, UV-Vis and fluorescence spectra were measured within three days after the S-S was extracted.

### 2.4. UV/Vis and Fluorescence Spectrophotometer Measurement

The filtered bulk P-W and L-W samples and the S-S samples were diluted with ultrapure water to a concentration of 5 mg-C/L to measure UV/Vis absorbance with a UV/Vis spectrophotometer (Hitachi, U-2900). The absorbance at 700–800 nm was used as the background; the absorbance of the sample was subtracted from the background value [15]. The samples were scanned at UV/Vis wavelengths 800–200 nm, and the scanning rate was 800 nm/min. The samples were acidified to a pH of less than 3 with 3 M sulfuric acid and analyzed by a fluorescence spectrometer (Hitachi, F-7000). Ultrapure water was used as a blank, and the measured data were subtracted from the blank. The UV/Vis absorbance at 254 nm was <0.2; therefore, the inner filter effect was ignored [21,31]. Excitation wavelengths were from 200 to 450 nm at 5 nm increments and emission wavelengths were from 250 to 550 nm at 2 nm increments. The scanning rate was 2400 nm/min.

### 2.5. Optical Index

The UV/Vis specific ultraviolet absorbance at 254 nm (SUVA_254_, L/mg-C/m) is calculated as Equation (1) [18]. The UV-Vis spectral slopes (S_275-295_) is the nonlinear fit of an exponential function to the absorption spectrum over the wavelength range between 275 and 295 nm [15]. The fluorescence index (FI) is calculated as Equation (2) [31]. The biological index (BIX) is calculated as Equation (3) [21].
(1)$$SUVA_{254}=A_{254}/\left[DOC\right]\times 100$$
(2)$$\mathrm{FI}\;=\frac{F_{\lambda em=450\;nm}}{F_{\lambda em=500\;nm}},\;\lambda_{ex=370\;nm}$$
(3)$$\mathrm{BIX}\;=\frac{F_{\lambda em=380\;nm}}{F_{\lambda em=430\;nm}},\;\lambda_{ex=310\;nm}$$
where [DOC] represents concentration of dissolved organic carbon (mg-C/L). A_254_ is the UV absorbance at 254 nm (cm^−1^). F_λem_ (380, 430, 450, 500 nm) are the fluorescent intensities at Emission wavelengths at 380, 430, 450 and 500 nm, respectively. λ_ex_ (310, 370 nm) are excitation wavelengths at 380 and 430 nm, respectively.

Three fluorescence EEM wavelength intensities were Ex/Em = 270 − 290/350 − 365 nm (PLF), Ex/Em = 320 − 340/400 − 415 nm (FLF) and Ex/Em = 370 − 390/460 − 475 nm (HLF) [26,27,32]. The PLF, FLF and HLF values are the sum of the fluorescence intensities of the measured solution in the design regions. The PLF, FLF and HLF organic carbon standardized (OCS) are the fluorescence intensity divided by DOC concentration for P-W, L-W and S-S samples.

### 2.6. Statistic

All statistical analyses were performed using S-Plus V 6.2 statistical software. Statistical significance was accepted at p < 0.05. Owing to the different size and lack of homogeneity of variance, the data sets were analyzed using a nonparametric test. Kruskal-Wallis tests were performed to compare the differences among the three data sets. Differences in the two data sets were analyzed using the Mann-Whitney *U* test. The EEM and UV/Vis indicators calculation and the EEM analysis and plots followed the R script developed by Lapworth and Kinniburgh [33]. This followed the needs of the experiments to modify the R software (2.13.2 V) script.

## 3. Results and Discussion

### 3.1. DOC Concentrations of DOM and S-S

The DOC concentrations of P-W, L-W and S-S are shown in Table 2. The average concentration of alkaline-extracted soil organic matter (S-S) (680 mg/kg) was higher than water extracted soil organic matter (WEOM) in agricultural soil (56–81 mg/kg) [16]. The DOC concentration of L-W DOM was significantly higher than P-W DOM (p < 0.001). In a soil column (30 cm high) infiltration test [34], the DOC concentrations were 0.80–1.01 mg/L and 1.14–2.38 mg/L in percolation and leachate water, respectively. The leachate DOC concentrations were higher than the percolation water DOC concentrations and the results were similar to the lysimeter test. However, the DOC concentration in groundwater was very low [6,29] and similar to P-W in the lysimeter test. In addition, the DOC concentrations of deep groundwater were much lower than DOC concentration in rainwater and surface water; the deep groundwater DOC concentration was significantly reduced compared to the surface water DOC concentration. For example, in the Yasu River watershed, Mostofa et al. [28] reported that the DOC concentrations were 1.31 ± 1.06 and 1.90 ± 0.74 mg/L for groundwater and river water, respectively. The DOC concentration of groundwater decreased. Shen et al. [6] reported that the DOC concentration of surface water was 9.47 ± 6.29 mg/L; the groundwater DOC concentration was significantly decreased, ranging from 0.82 to 1.19 mg/L.

### 3.2. Optical Indices

Optical methods have been widely applied to analyze the chemical composition and structure of DOM and S-S [11,15,18,20,35,36,37,38]. In this study, UV-Vis indices SUVA_254_ and spectral slope S_275-295_ and fluorescence indices FI and BIX were used to examine the chemical structure and composition of P-W, L-W DOM and S-S.

Table 2 lists the optical index values. The SUVA_254_ index is positively correlated with the aromatic content [11,18]. The quantitative order of average SUVA_254_ values was S-S > LW > PW, and the SUVA_254_ of S-S was significantly higher than the P-W value (p = 0.009). The SUVA_254_ value (<3 L/mg-C/m) indicated that the DOM solution mainly consisted of hydrophilic compounds, while the SUVA_254_ value (>4 L/mg-C/m) DOM solution mainly contained hydrophobic compounds [11]. The average SUVA_254_ values for the three solutions were 2.68, 3.37 and 3.84 L/mg-C/m for P-W, L-W and S-S, respectively. The SUVA_254_ values > 4 were 25, 13 and 38% for P-W, L-W and S-S, respectively, and the SUVA_254_ values < 3 were 63, 19 and 12% for P-W, L-W and S-S, respectively. The SUVA_254_ values suggested that the P-W DOM contained mainly hydrophilic compounds, L-W DOM and S-S contained partially hydrophilic and partially hydrophobic compounds [11,39].

The UV-Vis spectral slope S_275-295_ is inversely proportional to the average molecular weight [15,40], and the magnitude order of S_275-295_ was P-W > L-W > S-S, which suggested L-W had a lower average molecular weight than S-S, but had a higher average molecular weight than P-W.

Surface water SUVA_254_ values typically ranged from 1.0 L/mg-C/m to 6.0 L/mg-C/m [17]. Surface water had S_275-295_ values ranging from 0.014 to 0.018 nm^−1^ [15], and the terrestrial systems had values ranging from 0.012 to 0.023 nm^−1^ [41]. The DOM values of SUVA_254_ and S_275-295_ were comparable to those for surface water and deep groundwater, but the trend was reversed [4,6,30]. Shen et al. [6] investigated rainfall water and groundwater DOM. The results showed that the SUVA_254_ values were 2.88 ± 0.38 and 1.54 ± 0.09 L/mg-C/m, and the S_275-295_ values were 0.015 ± 0.001 and 0.021 ± 0.001 nm^−1^ for surface water and groundwater DOM, respectively. 

The hypothesis for surface water percolation into deep groundwater assumes that the DOM hydrophobic compounds sorb onto the minerals and partition into the soil organic matter [4,6,7]. Therefore, the deep percolation DOM contain a lower aromatic abundance and average molecular weight than the surface water DOM. The hypothesis is in contrast to the results of P-W and L-W in the lysimeter test.

FI provides an indicator for distinguishing DOM derived from terrestrial and microbial sources [20,36]. The low FI value (<1.4) had a strong terrestrial source contribution but the high FI value (>1.9) had a weak terrestrial source [31]. The BIX is an indicator for the presence of autochthonous material where a high value (>1.0) corresponds to a recently produced DOM of autochthonous origin. Low BIX value (<0.6) suggested an allochthonous origin [20,21]. The range of BIX and FI values of P-W and L-W showed that the DOM mainly contained biological or aquatic bacterial origin. S-S contained an intermediate autochthonous component.

The FI and BIX values of S-S were similar to the soil WEOM [16,42]. The FI and BIX values of P-W and L-W DOM were similar to the values of groundwater FI and BIX in rural areas [43]. The quantitative order was P-W > L-W > S-S for both FI and BIX values. S-S had the highest terrestrial and allochthonous sources but P-W had the highest aquatic and autochthonous sources. FI and BIX values of P-W DOM were significantly larger than L-W DOM (p < 0.01), suggesting that L-W DOM contained more terrestrial and less new microbial sources than P-W DOM [20,21].

### 3.3. Fluorescence EEM

The fluorescence excitation/emission matrices (EEMs), based on the Ex/Em maxima of the fluorescence peaks, are a useful tool for distinguishing different types and sources of natural waters and soil organic matter [23,24,38,44]. 

Figure 2a–c shows the fluorescence EEM of the three solutions that show P-W and L-W DOM had three major peaks (Peak B, Peak A and Peak C), and S-S had a fourth peak (Peak M). The Peak C excitation/emission wavelength was centered at Ex/Em = 320 − 325/400 − 425 nm, which is attributed to the UVC humic-like substances from terrestrial sources. The Peak A wavelength was centered at Ex/Em = 230 − 235/400 − 410 nm, which is attributed to the recently generated UVA humic-like substances. The Peak B wavelength was centered at Ex/Em = 220 − 230/300 − 310 nm, which is attributed to the tyrosine protein-like material. The Peak M wavelength was centered at the Ex/Em = 270/425 nm, which is classified as a marine humic-like substance, autochthonous production and a low molecular weight substance [14,20,23,25,44]. It noteworthy that the Peak C Em wavelength in the S-S and L-W was 10–15 nm right-shifted more than the P-W Em wavelength. 

The right shift of the Em wavelength represented a more conjugated structure and aromaticity in the L-W than P-W DOM, which can be attributed to the release and dissolution of the phenolic compounds from litter lignin in the S-S [5]. The Peak C Em right shift was similar to that reported by Mostofa et al. [28]. They were observed in the Yasu River watershed, Japan study, where the groundwater DOM (Ex/Em = 320 ± 9/425 ± 5 nm) and river water DOM (Ex/Em = 340 ± 5/432 ± 4 nm) were in contrast to the lysimeter results. The Yasu River water had a longer emission wavelength than the groundwater. The decrease in the Yasu River groundwater Em wavelength was attributed to the phenolic compounds of the river water DOM, which was adsorbed by soil minerals during river water infiltration. However, in the lysimeter study, the L-W DOM Peak C increased emission wavelength was attributed the SHS desorption of phenolic compounds that were dissolved into the L-W.

### 3.4. Protein- (PLF), Fulvic- (FLF) and Humic-Like (HLF) Fluorescence

The DOM and S-S compositions are a heterogeneous and complex mixture. The 3D fluorescence EEM provides the composition and structure information of fluorescent molecules. Different wavelength regions represent different fluorescent substances [20,23,24,25,44]. The values of the three EEM fluorescent regions PLF, FLF and HLF (Figure 2a–c) represent the fluorescence intensities of protein-, fulvic acid- and humic acid-like substances, respectively [26,27,45]. Changes in fluorescence intensity and ratios of PLF, FLF and HLF were used to track the DOM sources and composition changes [26,46,47].

Table 2 lists the total PFL, FLF and HLF fluorescent intensity of the three solutions as well as the OCS PFL, FLF and HLF (au/mg-C) fluorescent intensity. The total PLF, FLF and HLF fluorescent intensities were in the order of S-S > L-W > P-W, respectively. S-S contained the most abundant fluorescent substances, while the L-W DOM had a higher fluorescent intensity than the P-W DOM, which can be attributed to the fluorescent substances’ dissolution from S-S. Compared with P-W DOM, the L-W DOM average fluorescent intensities increased 115%, 141% and 181% for total PLF, FLF and HLF, respectively. The increased rates of total FLF and HLF were greater than the rate of the total PLF. Figure 3 shows the fluorescent distribution fractions of PLF, FLF and HLF for P-W, L-W and S-S, respectively. The DOM and S-S had the highest FLF percentage and the lowest HLF percentage. The S-S had the lowest PLF percentage but had the highest HLF percentage. The L-W increased the HLF fraction compared to the P-W (p = 0.079), which was attributed to the dissolution of the HLF from S-S.

The magnitude order of OCS PFL and OCS FLF fluorescence intensities was L-W > P-W > S-S, but the magnitude order of OCS HFL fluorescence intensity was L-W > S-S = P-W (Table 3). The OCS FLF and HLF fluorescence intensity of L-W was significantly greater than P-W (p < 0.001). Compared with the OCS P-W DOM, the OCS L-W DOM fluorescent intensity increased following the order HLF (81%) > FLF (55%) > PLF (44%). The total and OCS fluorescent intensities of L-W DOM were greater than P-W DOM for three fluorescent regions and increasing ratios of HLF and FLF were greater than PLF, which demonstrated that the L-W composition had higher humification substances than P-W.

Lapworth et al. [47] reported that the FLF/PLF ratios of DOM for two groundwater aquifers were 0.33–2.0 and 2.5–5.0, respectively. These two groundwater DOM were affected by sheep waste sources. The other DOM was affected by a terrestrial and microbial source mixture. The agricultural animal waste had high a PLF intensity; therefore, the FLF/PLF ratio was lower than the source of the terrestrial and microbial source mixture. Naden et al. [48] used FLF/PLF to distinguish DOM composition of different water bodies. The ratios of FLF/PLF of forest seepage, reservoir water and urban river DOM were 2.13, 0.90 and 1.45, respectively. The forest soil seepage contained more phenolic compounds due to soil dissolution, thus, the FLF/PLF ratio was higher than the other two DOM. The reservoir water had freshly produced DOM by algae and aquatic organisms and had the lowest FLF/PLF ratio. Baker [26] explored the FLF/PLF ratios of different river sections ranging from 0.65 to 3.3. The ratios of upstream and small stream were 1.0–3.33 (one exception 0.65), and the downstream ratios of a major river section were 0.83–1.43. The low ratios of the downstream section implied it received STP effluent with a high concentration of PLF, which reduced the FLF/PLF ratio.

Figure 4 shows the average ratios of FLF/PLF were 1.71 ± 0.78, 1.50 ± 0.18 and 1.96 ± 0.18 for P-W, L-W and S-S, respectively. The ratio of S-S was significantly greater than L-W and P-W (p < 0.001). The average ratios of HLF/PLF were 0.50 ± 0.24, 0.51 ± 0.07 and 0.90 ± 0.10 for P-W, L-W and S-S, respectively. The ratio of S-S was significantly greater than L-W and P-W (p < 0.001). The ratios of FLF/PLF and HLF/PLF between L-W and P-W were insignificantly different (p = 0.48 and p = 0.96, respectively). However, the coefficients of variance (VC) were 46% and 47% for the FLF/PLF and HLF/PLF ratios of P-W, respectively. The high VC values may not reflect the overall trend of the ratios.

The average values were further compared to the slopes of the linear regression. Figure 5a,b shows the linear regression between OCS FLF and OCS PLF as well as between OCS HLF and OCS PLF. The results showed that P-W, L-W and S-S had a significant linear relationship (except P-W OCS HLF and PLF linear regression had p = 0.046, the remaining five linear regressions had p < 0.01). The slopes of the linear regression were 0.62, 1.74 and 1.79 between OCS FLF and OCS PLF for P-W, L-W and S-S, respectively. The slopes of the linear regression were 0.15, 0.58 and 0.64 between OCS HLF and OCS PLF for P-W, L-W and S-S, respectively. The linear slopes of L-W were close to the S-S slopes and both slopes were greater than the P-W slope. The tendency of the linear regression slopes was different from the average values due to the large variation of PLF in P-W (Table 3). 

The linear regression slopes revealed that the chemical composition and structure of L-W were similar to S-S and greater than P-W DOM. The regression results were consistent with the results of the optic characteristics. For instance, the L-W DOM increased in aromaticity, average molecular weight and terrestrial source but decreased in the autochthonous sources. Additionally, the OCS FLF and HLF fluorescence intensity significantly increased compared to the P-W DOM. However, the average ratios of FLF/PLF and HLF/PLF did not reflect that trend. Hence, it is appropriate to use the linear regression slope instead of the average value when data have a high variation such as in the lysimeter case.

### 3.5. Percolation Mechanisms in the Lysimeter

The UV-Vis and fluorescence index results showed that the S-S had the highest extent of humification in the L-W and P-W DOM (the HLF fluorescence intensity and highest ratios of FLF/PLF and HLF/PLF). In addition, S-S had a slightly higher aromaticity, average molecular weight, terrestrial origin and allochthonous origin than the P-W and L-W DOM. In comparison with the optical characteristics of P-W DOM, L-W DOM increased DOC concentration, aromaticity (SUVA_254_), average molecular weight (S_275-295_) and terrestrial source contribution (FI decreases) but decreased the autochthonous DOM (BIX decreased). The Em wavelength of L-W DOM Peak C was 10 nm right-shifted compared to the P-W DOM Peak C. In addition, the OCS FLF and HLF fluorescence intensity of L-W DOM was significantly greater than the P-W DOM intensity. 

Generally, comparing surface water, groundwater DOC concentration decreased and the DOM microbial source increased [4,6,7]. The trend was in contrast to the L-W and P-W DOM optical characteristics in the lysimeter study. The hypothesis of deep vertical infiltration of surface water DOM was affected by the adsorption of soil minerals (aluminosilcate clay minerals and metal oxides/hydroxides in soil) and retained specific DOM functional groups [4,5,6,7]. DOM hydrophobicity component was partitioned into soil organic matter. Furthermore, the biodegradation of protein-like DOM by soil microorganisms produced the microbial metabolism. DOM hydrophobicity substances stayed on the mineral surface and were partitioned into soil organic matter, and the undecomposed protein-like DOM and soluble microbial byproducts infiltrated into the deep underground [4,6,7].

In this study, the increased DOC concentration and change in chemical composition and structure of the L-W DOM were attributed to the desorption of the S-S component during the P-W infiltration process. The results were in contrast to the results from the infiltration DOM into the deep groundwater because in the lysimeter test, percolation water was a shallow, fast and short-flowing lateral flow. The infiltration water DOM had a short contact time with the soil, and in the percolation process, the lignin phenolic compounds in the soil litter layer were desorbed and dissolved in the infiltration water DOM. With a short retention time, the desorption of S-S was superior to the DOM decomposition by microorganisms, which resulted in a higher DOC concentration of L-W than P-W. Rapid flow of infiltration water reduced soil adsorption/co-precipitation and metabolic breakdown by microorganisms, resulting in litter-derived DOM transportation. The L-W DOM Em wavelength right-shifted, showing that the L-W DOM had a more conjugated structure and aromatics, which attributed to S-S terrestrial and litter lignin phenolic compounds and dissolved humic acid-like substances [4,34,49]. The L-W OCS PLF, FLF and HLF fluorescence intensity increased, and the increase rate of fluorescence intensity was HLF > FLF > PLF. The average ratios of FLF/PLF and HLF/PLF showed that the P-W and L-W ratios were similar; however, the L-W linear regression slopes were similar to the S-S slopes and greater than the P-W slopes. These optical characteristics suggested that the shallow infiltration lateral seepage L-W DOM was affected by the S-S desorption mechanism. Since the soil depth of the lysimeter was 1.0 m and the collected drainage depth was 0.3 and 0.6 m each, it was a shallow lateral flow. The optical characteristics showed that the L-W DOM chemical composition and structural changes were caused by desorption of S-S. There is a significant difference between the shallow lateral flow and the deep vertical infiltration of groundwater DOM and S-S.

## 4. Conclusions

In this study, optical characteristics were used to investigate the chemical composition and structure of the DOM in the shallow infiltration lateral flow affected by soil S-S. The S-S had higher aromaticity, average molecular weight and terrestrial sources than the DOM. The L-W DOM had higher aromatic abundance (SUVA_254_) and average molecular weight (S_275-295_) than the P-W DOM. In addition, the L-W DOM had more terrestrial (Fl) but less autochthonous content (BIX) than the P-W DOM. The change in the chemical composition and structure in the L-W DOM was due to the desorption of S-S compounds during the infiltration process. The average ratios of FLF/PLF and HLF/PLF were insignificantly different between the P-W and L-W DOMs. However, linear regression showed that the slopes of the L-W DOM were much larger than the P-W DOM slopes, and the slopes of the L-W DOM were similar to the slopes of the S-S. The DOM composition and structure resulted from shallow P-W infiltration, which was different from the deep groundwater infiltration. The change in the L-W DOM in the shallow soil infiltration was substantial and was significantly affected by the S-S desorption.

## Figures and Tables

**Figure 1 sensors-19-04016-f001:**
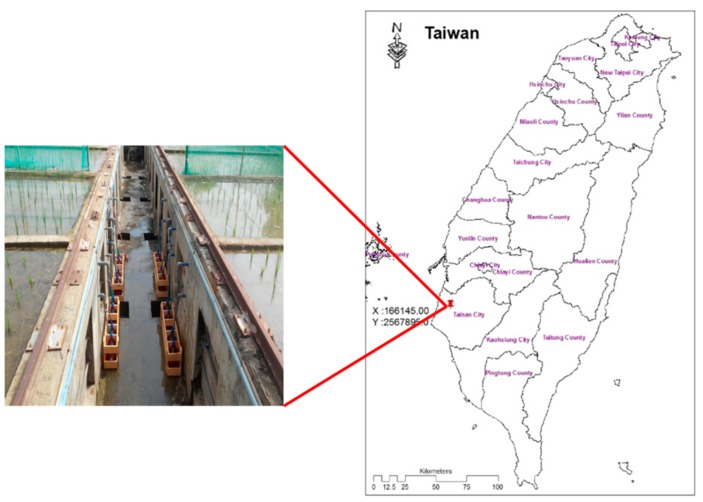
The configuration and placement of the experimental lysimeter.

**Figure 2 sensors-19-04016-f002:**
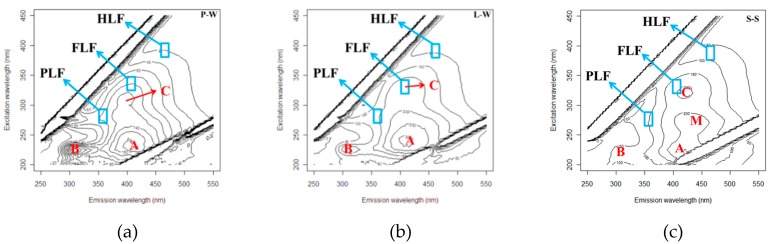
The excitation/emission matrix (EEM) plots of P-W, L-W and S-S, respectively. (**a**) P-W, (**b**) L-W, (**c**) S-S.

**Figure 3 sensors-19-04016-f003:**
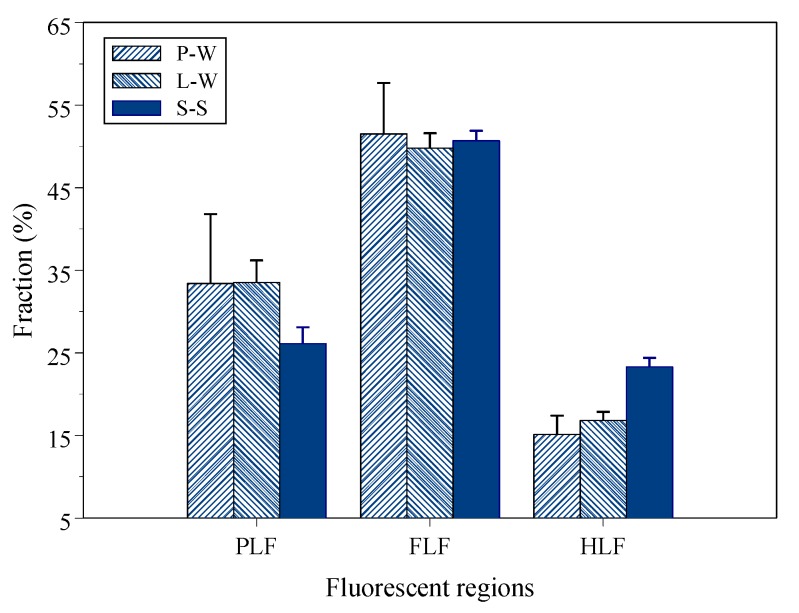
The PLF, FLF and HLF percentages for P-W, L-W and S-S.

**Figure 4 sensors-19-04016-f004:**
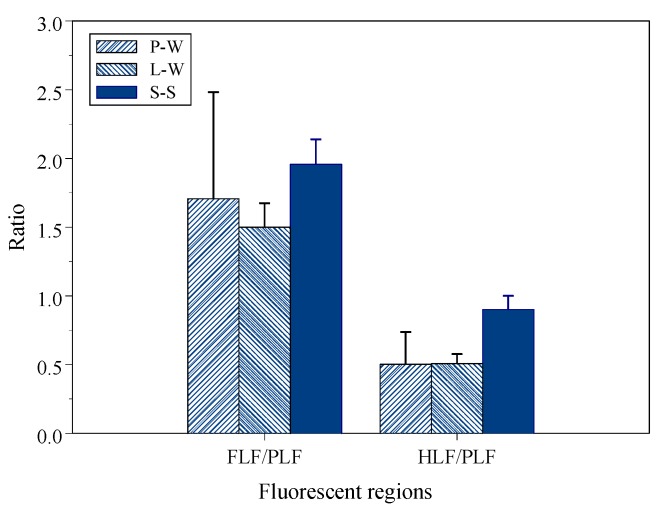
The OCS FLF/PLF and OCS HLF/PLF ratios for P-W, L-W and S-S.

**Figure 5 sensors-19-04016-f005:**
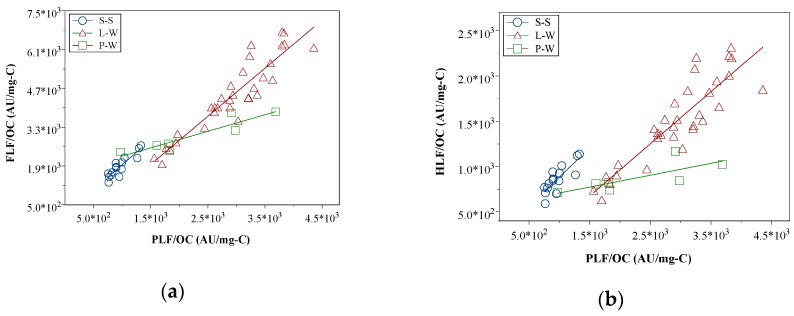
The linear regression of OCS FLF to OCS PLF (**a**) and OCS HLF to OCS PLF (**b**).

**Table 1 sensors-19-04016-t001:** Water quality of P-W and L-W and soil basic properties ^a^.

Sample	pH	EC μS/cm	DO mg/L	OM%
P-W	7.61 ± 0.42	1840 ± 240	1.44 ± 0.16	- *
L-W	7.90 ± 0.11	110.3 ± 5.8	-	-
S-S	8.24 ± 0.08	-	-	2.92 ± 0.37

* Did not measure; ^a^ sand = 45.8%, silt = 33.1% and clay = 21.1% (loam soil)

**Table 2 sensors-19-04016-t002:** Disolved organic matter (DOC) concentration and optical indices of P-W, L-W and S-S solutions.

	DOC *	SUVA_254_	S_275-295_	FI *	BIX *
	mg/L	L/mg-C/m	1/nm		
P-W	1.24 ± 0.35	2.68 ± 1.60	0.026 ± 0.011	1.94 ± 0.03	1.10 ± 0.07
L-W	2.01 ± 0.55	3.37 ± 0.69	0.019 ± 0.002	1.84 ± 0.05	0.98 ± 0.05
S-S	34.0 ± 11.0 ^a^	3.84 ± 0.55	0.011 ± 0.001	1.56 ± 0.05	0.71 ± 0.03

* P-W and L-W were significantly different (p < 0.05); ^a^ 680 ± 220 mg/kg.

**Table 3 sensors-19-04016-t003:** Total and organic carbon standardization (OCS) (in parentheses) fluorescent intensity of percolation water (P-W), leachate water (L-W) and soil solution (S-S) for the EEM three regions protein- (PLF), fulvic- (FLF) and humic-like fluorescence (HLF).

	PLF (au)	FLF (au)	HLF (au)
P-W	2640 ± 1482 (2048 ± 1064)	3583 ± 1259 (2868 ± 697)	1042 ± 365 (833 ± 188)
L-W	5687 ± 1389 (2940 ± 710)	8642 ± 2937 (4456 ± 1335)	2931 ± 1067 (1505 ± 462)
S-S	32,128 ± 7646 (966 ± 193)	61,901 ± 10,632 (1886 ± 385)	28,520 ± 5480 (862 ± 152)

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
