# Peer review of "Fluorescence Characteristics of Dissolved Organic Matter (DOM) in Percolation Water and Lateral Seepage Affected by Soil Solution (S-S) in a Lysimeter Test"

_sensors, 2019, doi:10.3390/s19184016_

Round 1

Reviewer 1 Report

In the present manuscript, the authors compare the UV and fluorescence properties of percolation water (P-W) and leachate water (L-W) DOMs affected by soil humic substances in order to examine the interaction between them. The idea and the design of the sampling are good. I am disappointed by some part of the manuscript. The quality of the manuscript is not equal in all parts (please refer to my general and specific comments). I won’t reject the manuscript, however, I expect a intensive review of the current version before acceptance.

General comments:

1). My first concern is related to the quality of the manuscript and its structure. Several times, new results come up without any previous introduction in the M&M. Readers have to go back and forth along with the manuscript as it is messy and disorganized. Some parts scientifically look quite weak leading me to believe the authors do not completely master their topic. The whole manuscript needs to be reviewed.

2). My second concern is regarding the terminology used for the soil extract. The authors called it Soil Humic Substances (SHS). Although it has been demonstrated that alkaline extraction preferentially extracts Humic Substances, the soil extract solution does not strictly consist of Humic Substances. I suggest to change it for something more accurate. “Soil extract” or “Soil solution” would be more appropriate.

3). My last one is regarding the fluorescence data. The authors basically did peak picking. Nowadays, when we deal with fluorescence data we combine the measurement with PARAFAC analysis to generate fluorescent components. These components can, later, be compared in OpenFluor database for assignment. This approach will give more strength to the study and also refine the assignment did in part 3.3.

However, as the discussion is based on studies handling the fluorescence data in the same way, I would not request to use PARAFAC analysis for this study. But I would highly encourage the authors to consider this option for their next work.

Specific comments:

Line 43-46: The sentence is too long.

Line 47: What does that mean? DOM biological??.

Line 48: replace “yields” with “yield”.

Line 49: phenolic compounds. Reference?

Line 32-81: the introduction needs to be deeply revised. I think most of the information we can expect in an introduction is here, but the structure is not well organized and paragraphs are not well linked to each other.

Line 110: 0.841mm. Seriously?

Line 113: replace “Slurry” with “suspension”.

Line 116: “as soon as possible”. This is not rigorous. Re-write it.              

Line 124: Why do the authors change the pH at 3? Do they know that changes in pH affect the fluorescence properties of the DOM?

Lines 123-126: This part needs to be re-write. It is quite messy. It starts with the correction applied to the spectra before explaining the conditions to obtain spectra. Then, it seems they did not do the Raman normalization. Why?

Lines 130-136: It would be better to write the equation related to the indices instead of all this text. It will easier for the reader.

Line 153-156: What is the link between this information and the DOC? And why do the authors talk about forest soil and agricultural soil? In this respect, what are the characteristics of the soil used in this study?

Line 169: Table 1, Wasn’t it possible to have a concentration for SHS in mg/ml, was it? Probably you have as you calculate SUVA?

Table 1: FIX? No, FI.

Line 165: Which groundwater? Which river?

Line 178: Replace ‘SUW’ by ‘SUVA’

Line 185: What? Where does this conclusion come from? What is the link between SUVA (aromaticity index) and the hydrophilic property?

Line 197: Any hypothesis to explain this result?

Line 200: Re-write.

Line 202: Autochthones?

Line 207: Why do not compare with soil alkaline extraction?

Line 228-237: Again here, the authors talked about river water, groundwater, P-W, and L-W. It is confusing.

Line 239: It could also have been done with the component identified by PARAFAC analysis.

Line 254: it would be more accurate to say distribution than intensity as it is illustrated in percent in figure 3.

Line 289-300: And then? There are only observations here.

Line 298-299: remove it.

Line 299-300: justify it.

Line 304 and 306 (Figure 3 and 4); change ‘species’ by ‘fluorescent regions’.

Line 312: replace ‘allochthones’ with ‘allochthonous’

Line 349: replace ‘change’ with ‘changes’

Reviewer 2 Report

The manuscript “Fluorescence characteristics of dissolved organic matter (DOM) in percolation water and lateral seepage affected by soil humic substances (SHS) in a lysimeter test” by Chiu Teng-Pao et al. deals with the study of chemical structure of dissolved organic matter, as percolation water and leachate water, and soil humic substances from a lysimeter by means of optical indices and fluorescence spectroscopy. The topic is of interest and fits within the aims of the Journal. The experimental design is fine and the results are new. I think minor changes should be necessary to the manuscript before it can be considered ready for publication.

I think of interest should be if the authors may add a table with the soil characteristics of the trial.

Lines 94-96. Where are these data?

Line 113. Change rpm in x g.

Lines 144-145. Pearson is generally used for calculating the matrix of correlation. Are you used linear regression with Pearson correlation?

Lines 322-323. Data not showed. Add these data or change the sentence by a hypothesis.

Lines 55-59. Not necessary to repeat the same references. In addition to these characteristics, the authors should also mention other analyzes that qualify the dissolved organic substance. See for example the works by Pizzeghello et al. 2006 (Chemosphere 65(2), pp. 190-200) and Nardi et al. 2003 (Journal of Chemical Ecology 29(7), pp. 1549-1564).

Round 2

Reviewer 1 Report

Few minor comments.

Line 102: "did not measure" refers to?

Line 111: "sieved with 2 nm"? You mean "sieved at 2.0 mm"? Right?

Line 111: what does "RSG" mean?

Line 114: add S-S after soil solution or remove "soil solution".

Line 126-127: add a reference to support the value of 0.2.

Equation 3: The police for the emission wavelength should be all in subscript.

Line 167: remove "Japan study".

Line 182-185: "The SUVA254 value (< 3 L/mg-C/m) indicated that the DOM solution mainly consisted of hydrophilic compounds, while the SUVA254 value (> 4 L/mg-C/m) DOM solution mainly contained hydrophobic compounds". Add a reference. I would be curious to see the paper relating that.

Line 207: I am not convinced by "metric". Please replace it.

Figure 3: the x-axis disappears where the bars with a pattern are. 
